Selecting appropriate variables for detecting grassland to cropland changes using high resolution satellite data

Klouček Tomáš tkloucek@fzp.czu.cz 1
Moravec David 1
Komárek Jan 1
Lagner Ondřej 1
Štych Přemysl 2
1 Department of Applied Geoinformatics and Spatial Planning, Faculty of Environmental Sciences, Czech University of Life Sciences Prague , Prague , Czech Republic
2 Department of Applied Geoinformatics and Cartography, Faculty of Science, Charles University , Prague , Czech Republic
Yu Le
Electronic publication date: 2018 Sep 6
Publication date: 2018
Volume: 6
Electronic Location ID: e5487
Received 2018 Feb 27; Accepted 2018 Jul 30
Copyright: ©2018 Klouček et al.
Copyright year: 2018
Copyright holder: Klouček et al.
License: This is an open access article distributed under the terms of the Creative Commons Attribution License, which permits unrestricted use, distribution, reproduction and adaptation in any medium and for any purpose provided that it is properly attributed. For attribution, the original author(s), title, publication source (PeerJ) and either DOI or URL of the article must be cited.
License URL: https://creativecommons.org/licenses/by/4.0/

Keywords: Change detection (CD), Grassland, Tasseled Cap (TC), Cropland, Normalized Difference Vegetation Index (NDVI), Variables

Funding: Czech University of Life Sciences Prague (CULS) 20174208 This work was supported by the Czech University of Life Sciences Prague (CULS) under grant no. 20174208. The funders had no role in study design, data collection and analysis, decision to publish, or preparation of the manuscript.

==============================
Grassland is one of the most represented, while at the same time, ecologically endangered, land cover categories in the European Union. In view of the global climate change, detecting its change is growing in importance from both an environmental and a socio-economic point of view. A well-recognised tool for Land Use and Land Cover (LULC) Change Detection (CD), including grassland changes, is Remote Sensing (RS). An important aspect affecting the accuracy of change detection is finding the optimal indicators of LULC changes (i.e., variables). Inappropriately selected variables can produce inaccurate results burdened with a number of uncertainties. The aim of our study is to find the most suitable variables for the detection of grassland to cropland change, based on a pair of high resolution images acquired by the Landsat 8 satellite and from the vector database Land Parcel Identification System (LPIS). In total, 59 variables were used to create models using Generalised Linear Models (GLM), the quality of which was verified through multi-temporal object-based change detection. Satisfactory accuracy for the detection of grassland to cropland change was achieved using all of the statistically identified models. However, a three-variable model can be recommended for practical use, namely by combining the Normalised Difference Vegetation Index (NDVI), Wetness and Fifth components of Tasselled Cap. Increasing number of variables did not significantly improve the accuracy of detection, but rather complicated the interpretation of the results and was less accurate than detection based on the original Landsat 8 images. The results obtained using these three variables are applicable in landscape management, agriculture, subsidy policy, or in updating existing LULC databases. Further research implementing these variables in combination with spatial data obtained by other RS techniques is needed.

Introduction

Land Use and Land Cover (LULC) techniques form an integral part of many studies (Kindu et al., 2013; Gupta & Shukla, 2016; Chaudhuri & Mishra, 2016) overlapping with other research fields (Cardinale et al., 2012). LULC is considered an important factor influencing the environment and its changes have a demonstrable impact on climate change (Tasser, Leitinger & Tappeiner, 2017). Among the land cover types in the European Union (EU), grassland and cropland are the most prominent, accounting for 44% of the total area (Eurostat, 2017). Since the 1990s, the main LULC change trends in most post-communist Central European countries are afforestation, grassing over, intensification, and urbanisation. Even though the change of grassland to cropland is not as frequent a transition as it was during the communist era (Kupková & Bičík, 2016), it still elicits a significant impact on the ecosystem. Grassland plays an irreplaceable role as a natural habitat of many organisms, helps with the accumulation of greenhouse gases, prevents erosion, keeps water in the landscape and reduces pollution (European Union, 2016). However, these benefits are easily disrupted by ploughing the grassland, thus turning it into cropland. It is, therefore, important to detect such changes, quantify them and continuously monitor the developments. The occurrence of new cropland at the expense of grassland is especially prominent in post-communist states that have recently joined the EU and started to receive agricultural subsidies (Pazúr et al., 2014). This process is also affected by a number of national and European agricultural policies and initiatives (Sklenicka et al., 2014), such as the Good Agricultural and Environmental Conditions (GAEC) (Sklenicka et al., 2015). Change data acquired from remote sensing based models can, therefore, serve both as a basis for decision-making in the landscape management and have a socio-economic application in agriculture and its subsidy policy (Esch et al., 2014).

The primary data source for LULC Change Detection (CD) is Remote Sensing (RS). Multi-spectral satellite images are one of the most commonly used types of RS data, among which Landsat satellites images stand out due to long-term imaging, a suitable compromise between spectral, spatial and temporal resolution and free availability (Wulder et al., 2008; Xian, Homer & Fry, 2009; Chen et al., 2012; Roy, Ghosh & Ghosh, 2014). LULC change detection using RS data is based on the theoretical assumption that each LULC type has its own typical spectral signatures. If an LULC type changes, so will its spectral signatures (Hussain et al., 2013). In practice, it is often difficult to distinguish the signal of true changes from the false signals arising from external factors (different atmospheric conditions, soil moisture, or the phenological stage) (Jensen, 1996), the selection of RS data (Lu, Li & Moran, 2014), pre-processing (Dai, 1998) and atmospheric corrections (Song et al., 2001), the choice of the change detection method, the selection of the variables or the inexperience of the analyst (Lu et al., 2003). The significance of these uncertainties is even greater in LULC objects with very similar spectral signatures, which is exactly the case in croplands with a high degree of heterogeneity and significant effects of different phenological phases of individual crops and plants (Lu et al., 2003).

Some studies dealing with the classification and change detection of grassland and cropland have been published (Chen & Rao, 2008; Esch et al., 2014). These categories are often a part of a comprehensive change detection study (Mas, 1999; Bergen et al., 2005; Wondrade, Dick & Tveite, 2014; Vorovencii, 2014). We can also find studies aimed at a more detailed classification on the level of individual croplands (Wardlow, Egbert & Kastens, 2007; Turker & Ozdarici, 2011) or on grassland change detection (Weeks et al., 2013). Studies focusing specifically on grassland to cropland change are, however, still exceedingly rare (Tarantino et al., 2016). Among the studies closest to the topic of our study, the papers by Tarantino et al. (2016), who achieved 86.91% accuracy in the detection of semi-natural grassland to cropland changes in Italy using a cross-correlation analysis of Landsat 8 OLI images, and by Weeks et al. (2013), who used NDVI differencing for the change of “indigenous” grasslands in New Zealand and achieved 56% accuracy, can be mentioned.

Many papers have been published that reviewed the methods and techniques used for the detection of LULC changes (Singh, 1989; Lyon et al., 1998; Lu et al., 2003; Coppin et al., 2004; Berberoglu & Akin, 2009; Bhandari, Kumar & Singh, 2012; Hussain et al., 2013; Lu, Li & Moran, 2014; Tewkesbury et al., 2015), in forest ecosystems (Coppin & Bauer, 1996; Woodcock et al., 2001; Lu, Batistella & Moran, 2008), urban areas for building detection (Liu & Zhou, 2004; Sohn & Dowman, 2007; Aleksandrowicz et al., 2014) or for the detection of imperious surfaces (Xian, Homer & Fry, 2009). Other studies focus on the problem of mapping the general land use change (Yin et al., 2014) or on agricultural land specifically (Weeks et al., 2013; Müller et al., 2015; Tarantino et al., 2016). The application of RS in agriculture is summarised, for example, in a review by Atzberger (2013). The current trend uses a time series for agricultural change detection (for example, all the available Landsat imagery), which provides additional phenological information (Müller et al., 2015). In many cases, an insufficient number of satellite images is available due to cloud cover and, therefore, bi-temporal change detection is still needed. The alternative approach uses imagery from two dates, for which the time of the acquisition and the variable selection are crucial. The potential usefulness of various CD variables and their impact on LULC CDs has not been sufficiently studied either.

Variables used for CD may be divided into three categories. One category consists of spectral variables that include spectral bands and derived vegetation indices, transformed images, segments, sub-pixel features, and classification results. The second category includes spatial variables such as textures, different scales, the complexity of the landscape or topography. The temporal variables comprise the third category (Lu, Li & Moran, 2014). With more than 40 modifications, vegetation indices form the most numerous group of variables (Bannari et al., 1995). Significant variability and the amount of RS data, as well as the choice of variables, are very likely to affect the LULC CD, as was shown in other spatial analyses (Barry & Elith, 2006; Moudrý & Šímová, 2012; Klouček, Lagner & Šímová, 2015). Using a large number of variables can potentially improve the accuracy of the CD. On the other hand, such an approach can introduce a number of uncertainties into the detection and make the interpretation of obtained results difficult (Lu & Weng, 2007).

Despite the fact that LULC change detection has been one of the most discussed RS topics for decades, to the best of our knowledge, few studies have focused their attention on selection of appropriate variables for detection of changes in croplands. The aim of our study is to find the optimal variable(s) for grassland to cropland detection based on the Landsat 8 OLI high resolution data and the vector database, called the Land Parcel Identification System (LPIS), and to test the results for the 2013-2016 period on the selected territory. We hypothesised that (1) it is possible to find a suitable variable or group of variables capturing the change of the grassland to cropland due to different spectral profiles; (2) the greater the amount of the incorporated variables, the more accurate the CD would be; (3) spectral variables would be more significant than textural ones; (4) an important aspect of the grassland to cropland change detection would be the time of the acquisition input satellite data.

Materials and Methods

Study area

The study area is located in Central Europe, namely in the western part of the Czech Republic intersecting with Landsat 8 scene No. 192/25 with centre point coordinates approximately 50°22′N, 13°41′E, see Fig. 1. The study area is on a regional scale (approx. 36,260 km2) and is characterised by notable variability (topographical, landscape ecology as well as vegetational variability). This scale and localisation therefore warrants the occurrence of a sufficient number of both grassland to cropland changes and of no-change areas. The expected occurrence of changes was manually verified prior to the analysis using freely available CORINE Land Cover data (http://land.copernicus.eu/pan-european/corine-land-cover/).

Figure 1 The study area is (located in the Czech Republic, specifically) comprising a part of Landsat 8 scene Path 192 Row 25.

Input data

The main data source was a pair of high resolution images taken by the Landsat 8 OLI on August 3rd, 2013 and August 27th, 2016. The images downloaded from the US Geological Survey (http://earthexplorer.usgs.gov/) contain nine spectral bands with a resolution of 30 m (multi-spectral) and 15 m (panchromatic), respectively. Detailed specifications of the OLI sensor can be found in Roy et al. (2014). At the time of the image selection, the chosen images were the only ones available for a pair of scenes that, besides being almost cloudless, also met the other criteria including the suitable extent, the sufficient temporal distance between the imaging data, and acquisition at the suitable phenological stage. The most suitable period for the grassland to cropland change detection is the period shortly after harvest (late summer, early autumn) (Esch et al., 2014).

As a source of reference data on the use of the agricultural land, we used the Land Parcel Identification System and its vector database containing the land use data for the entire territory of the Czech Republic from 2004. The basic unit of LPIS is a group of adjacent plots representing a continuous area farmed by a single farmer with a single crop plant. The database classifies the agricultural land into 11 land use categories. Data from years corresponding with the Landsat images, i.e., 2013 and 2016, were used, see Fig. 2. In accordance with LPIS classification, cropland is defined as a “farmed land producing crop plants requiring annual replanting, which is not grassland” in this study. Grassland, on the other hand, is defined as a “farmed land under permanent pasture or, where appropriate, contiguous vegetation dominated by grass, used predominantly for feeding or technical purposes” (The Ministry of Agriculture of the Czech Republic, 2016).

Figure 2 An example of used datasets. Landsat 8 images, NDVI vegetation index, and (no-)change grassland to cropland plots (LPIS database) from 2013 and 2016.

(A) Landsat 8 image from 2013. (B) Landsat 8 image from 2016. (C) NDVI RGB composite (R = NDVI 2013, G = NDVI 2016, B = NDVI 2013). (D) (No-)change grassland to cropland plots from LPIS database.

Images and data pre-processing

Landsat 8 OLI images were obtained at a Level-1T processing level, which includes standard radiometric, geometric and terrain correction using Ground Control Points and the Digital Elevation Model. The results of this step were visually inspected for accuracy with regard to the geometric overlay of the images and the LPIS database. No additional image to image registration was needed. The raw Digital Number data was converted to surface reflectance (Song et al., 2001) using FLAASH (Fast Line-of-sight Atmospheric Analysis of Hypercubes) in ENVI software (version 5.4), and any areas obscured by clouds were manually removed from the image.

From the LPIS database, both plots with grassland to cropland change and those on which the grassland remained were extracted. Plots detected as croplands in both time points (information acquired from LPIS also) were removed from the calculation. In the area of interest, 570 changed LPIS plots and 33,196 no-change LPIS plots were identified. To minimise the mixed pixel effect, only plots larger than 1 hectare with a non-elongated shape were selected. A non-elongated shape was defined as the proportion between the shape area (ha) and the shape length (m), which had to be greater than 0.045. This threshold value was expertly set based on the visual inspection and knowledge of the LPIS database. On the acquired sample, a visual check that focused on the homogeneity of the selected plots was carried out based on the freely available orthophotos of the Czech Republic. See Fig. 3 for data processing workflow.

Figure 3 A scheme of the study methods describing data processing workflow.

For validation of models multi-temporal change detection based on object-based classification using Support Vector Machine algorithm was used.

Selection and calculation of the variables

For each scene, 59 LULC change detection variables were calculated. Specifically, the calculated variables included 36 vegetation indices, 10 textural characteristics, seven components of Principal Component Analysis, and six Tasselled Cap components (Table 1). The numbers of variables represent, in our opinion, potentially used spectral and spatial indicators for change detection in the ENVI software by a common user. The calculation of the variables was performed by algorithms implemented in ENVI. Spectral-based variables were calculated from pre-processed spectral bands, while textural variables were calculated from the panchromatic band (see ENVI help in Table 1). For each variable, the mean value for every plot of the LPIS-acquired database was obtained using the ArcGIS (version 10.4) Zonal Statistics tool for both 2013 and 2016.

Table 1 Fifty-nine change detection variables used in the study for detection of (no-)change from grassland to cropland.

Specifically, 36 vegetation indices, 10 texture characteristics, seven components of Principal Component Analysis and six components of Tasseled Cap were used. Numbers represent almost all available variables in ENVI software. For details see external links.

Group	Change detection variables	
Vegetation Indices	Atmospherically Resistant Vegetation Index, Burn Area Index, Clay Minerals, Difference Vegetation Index, Enhanced Vegetation Index, Ferrous Minerals, Global Environmental Monitoring Index, Green Atmospherically Resistant Index, Green Difference Vegetation Index, Green Normalized Difference Vegetation Index, Green Ratio Vegetation Index, Green Vegetation Index, Infrared Percentage Vegetation Index, Iron Oxide, Leaf Area Index, Modified Non Linear Index, Modified Normalized Difference Water Index, Modified Simple Ratio, Modified Triangular Vegetation Index, Modified Triangular Vegetation Index, Improved Non-Linear Index, Normalized Burn Ratio, Normalized Difference Built Up Index, Normalized Difference Snow Index, Normalized Difference Vegetation Index, Optimized Soil Adjusted Vegetation Index, Red Green Ratio Index, Renormalized Difference Vegetation Index, Simple Ratio, Soil Adjusted Vegetation Index, Structure Insensitive Pigment Index, Sum Green Index, Transformed Difference Vegetation Index, Visible Atmospherically Resistant Index, WorldView Improved Vegetative Index, WorldView Water Index	
Texture	Contrast, Correlation, Data Range, Dissimilarity, Entropy, Homogeneity, Mean, Skewness, Second Moment, Variance	
Principal Component Analysis	PCA 1, PCA 2, PCA 3, PCA 4, PCA 5, PCA 6, PCA 7	
Tasseled Cap	Brightness, Greenness, Wetness, Fourth, Fifth, Sixth	
Notes.

For more information about the variables visit http://www.harrisgeospatial.com/docs/alphabeticallistspectralindices.html or http://www.harrisgeospatial.com/docs/backgroundtexturemetrics.html.

Statistical assessment

To determine the optimal set of variables for grassland to cropland change detection, we first excluded the highly correlated ones (r > 0.9) from the full correlation matrix (see Supplemental Information 1). Where correlations were detected, only the variable most frequently used in the available literature was included into the subsequent analysis. From the original set of 59 variables, 41 were eliminated in preselection due to high correlation and the uncorrelated variables are presented in Table 2.

Table 2 Non-correlated variables used for detecting grassland to cropland (no-)changes.

Group	Not correlated variables	
Vegetation indices	Normalized Difference Vegetation Index, Simple Ratio, Sum Green Index	
Texture	Contrast, Data Range, Entropy, Homogenity, Mean, Second Moment, Skewness	
Principal component analysis	PCA 1, PCA 2, PCA 3, PCA 4, PCA 7	
Tasseled cap	Brightness, Wetness, Fifth	

The best set of variables was found using logistic regression specifically based on the lowest AIC (Akaike Information Criterion) (deLeeuw, 1992) using Generalised Linear Models (GLM) with a defined binominal distribution of errors (more about GLM can be found, e.g., in Dobson & Barnett, 2008). Models, from one to seven members, were found by permutation of all the combinations of variables with the ‘glmulti’ package in R (version 3.3.2). Models with a higher number of variables than seven were best found by AIC in a Stepwise Algorithm in R because of the time-consuming nature of the previous method. The calculated AIC values for the models based on two - 14 variables were very similar (only one-variable model using AIC values was significantly different), so only the models, where the AIC values are at least slightly changed (one, three, five, seven, 14), were chosen for the accuracy assessment.

Classification and accuracy assessment

A practical accuracy assessment of the created models and the Landsat 8 images only (Table 3) was undertaken using the object-based multi-temporal change detection. The variables of the models from both years were merged, based on statistic calculation, into a single image (Layer stacking tool). The training data for classification was selected from all of the 33,766 plots from pre-prepared LPIS database (‘Images and data pre-processing’). Based on stratified random sample design, 300 plots with change and 1200 without change were chosen (Congalton & Green, 2009). Borders of selected plots from LPIS database were used as the segments of the object-based classification. Using slides consisting of variables and training data, change maps were created in ENVI software. Due to non-normal distribution of the input data, the non-parametric Support Vector Machine (SVM) classifier (Lu & Weng, 2007) was used for classification. The settings of the SVM algorithm was set as the default. The Kernel type: Radial Basic Function; Gamma in Kernel Function: the inverse of the number of bands in the input image; The Penalty Parameter: 100; The Pyramid Levels: 0; and the Classification Probability Threshold: 0. The same methodology was used for the change detection based only on the Landsat 8 images (the amount of training and validation samples, classification algorithm, etc.).

Table 3 Summary of the validated models for the grassland to cropland change detection based on different set of variables.

The value of AIC specifies the information potential of models.

No. of variables	Change detection model	AICa	
One	Normalized Difference Vegetation Index	5,633.39	
Three	Normalized Difference Vegetation Index, Wetness, Fifth	4,592.41	
Five	Normalized Difference Vegetation Index, Wetness, Fifth, Brightness, Sum Green Index	4,263.74	
Seven	Normalized Difference Vegetation Index, Wetness, Fifth, Brightness, Sum Green Index, Second Moment, PCA 2	4,060.35	
Fourteen	Normalized Difference Vegetation Index, Wetness, Fifth, Brightness, Sum Green Index, Second Moment, PCA 2, PCA 1, PCA 3, PCA 4, PCA 7, Data Range, Contrast, Skewness	3,950.90	
Notes.

a AIC (Akaike Information Criterion).

Finally, the accuracy of the change maps was calculated by comparison with stratified random validation (testing) samples extracted from the pre-prepared LPIS database (excluding the training data) using an confusion matrix. The sampling design was inspired by Zhen et al. (2013) and Olofsson et al. (2014). The assessment was based on evaluating the number of correctly classified 200 change and 800 no-change plots into change maps with validation plots from the LPIS database. A 95% confidence interval was calculated from the overall accuracy of the models. The models, accuracy has been tested with a homogeneity test of binominal distribution. The models have been tested against each other using Holm’s p-value adjustment for multiple comparisons.

Results

Models for change detection

The lowest AIC was obtained from the model with 14 variables (3950.90), the highest from the model using a single variable (5633.39). The single most significant variable was the NDVI (Normalised Difference Vegetation Index), which was represented in all the models. In the models with a lower number of variables, variables based on spectral information were predominantly used. The separability of the model with one variable (NDVI) is demonstrated by Fig. 4. With additional variables, textural variables began to play a greater role, see Table 3. The summary of calculated models can be found in Supplemental Information 2.

Figure 4 2D scatter plot created from NDVI average values of change and no-change plots.

Points represent training data (300 change, 1,200 no-change plots). X-axis belongs to NDVI 2016 and Y-axis belongs to NDVI 2013 (one-variable model).

Change maps evaluation

The overall accuracy of the change maps generally increases with the increasing number of variables in the models. The best change map was created from the highest number of variables (89.80% accuracy, Kappa 0.63), however classification based on a single variable provided only slightly inferior results (88.10% accuracy, Kappa 0.55) as illustrated in Table 4. These findings were statistically confirmed by the homogeneity test for binominal distribution. So, we cannot conclude (on a 95% confidence level), that one of the models is more accurate, see Fig. 5.

Table 4 The accuracy of models (%) calculated based on different sets of variables by non-parametric classifiers Support Vector Machine (SVM).

No. of variables/model	Change PA	No-change PA	Change UA	No-change UA	OA	95% CI	
One	46.00	98.63	89.32	87.96	88.10	86.09–90.11	
Three	49.50	98.88	91.67	88.68	89.00	87.07–90.94	
Five	46.50	99.00	92.08	88.10	88.50	86.52–90.48	
Seven	52.00	98.25	88.14	89.12	89.00	87.06–90.94	
Fourteen	55.50	98.38	89.52	89.84	89.80	87.93–91.68	
Landsat image	59.00	98.25	89.39	90.55	90.40	88.57–92.23	

Figure 5 Overall accuracy (%) of calculated models with 95% confidence intervals.

Figure 6 Comparison of created change maps with Landsat 8 images and LPIS database.

(A) One-variable model. (B) Three-variable model. (C) Fourteen-variable model. (D) Landsat 8 images only model. (E) Landsat 8 image from 2013. (F) Landsat 8 image from 2018 with (no-)change plots from LPIS database.

Looking more closely, the improvement in accuracy with an increasing number of variables is associated only with the increasing Producer’s Accuracy (PA) of the change class (one-variable model 46.00% and fourteen-variable model 55.50%). As shown in Table 4, there is an improvement in the change class PA quality of the model between the models using one and three variables. The rest of the confusion matrix parameters (User’s Accuracy, Commission and Omission) were very similar in all the cases. Contrary, the no-change detection did not show any notable improvement with an increasing number of variables (PA 98.25–99.00%). All change maps, however, underestimated the number of change plots and overestimated the number of grassland to cropland no-change plots (Fig. 6). The results indicate that classification of the change and no-change plots has achieved sufficient accuracy. If we compare the accuracy of the change maps based on a statistically selected set of variables with change maps created from the Landsat images (OA 90.40%, Kappa 0.66), there is not any significant difference. The detailed confusion matrices are available in Supplemental Information 3.

Discussion

In accordance with the results, it is possible to use statistically selected variables for detection of grassland to cropland land cover changes. At first sight, it could be apparent that it is sufficient to only use the NDVI vegetation index for this type of analysis. However, based on the visual inspection of the misclassification in all the change maps and the confusion matrix (Supplemental Information 3), it is clear that the largest change detection inaccuracy is in a case when differentiating grassland and cropland plots with green plants. The largest number of these plots were poorly classified in the case of using only a one-variable model based on NDVI (the lowest Producer’s Accuracy). This result is not surprising because the surface reflectance of both categories is, in the spectral range of the Landsat 8 bands, almost identical and the NDVI index even uses two spectral bands (Red and Near Infrared). Only the NDVI variable can be used in the situation, when almost all plots are in the same phenological phase. However, this is not the case of our study and it is not common in the most of analyses, where some parts of the area (mountains vs. lowlands) are in different phenological phases. Therefore, the addition of some variables based on another spectral band is needed.

In our study, almost all vegetation indices were significantly correlated. The NDVI variable was chosen as the most appropriate because of its frequency of use in research. The statistical evaluation, however, indicates that very similar results would be achieved with any of the other vegetation indices closely correlated with the NDVI one, see the correlation matrices in Supplemental Information 1.

A good compromise among improving the accuracy of detection, the demands for computational time and complications of the interpretation of the obtained results, seems to be supplied by NDVI with the Wetness and Fifth components of Tasselled Cap (three-variable model in the study). These variables are more sensitive to different conditions of the grassland plots and cropland plots with the green plants. The advantage of the three-variable model is also the relatively small number of variables, allowing the utilisation of methods based on the determination of an optimal change detection threshold (Chen & Rao, 2008; Otukei & Blaschke, 2010). These findings related to crop phenology, besides other conclusions, point an importance of appropriate time acquisition of satellite images. It also confirms the hypothesis about an importance of this aspect for the grassland to cropland change detection.

The suitability of NDVI for the classification and change detection has been demonstrated in several studies (Lunetta et al., 2006; Wardlow, Egbert & Kastens, 2007; Pu et al., 2008; Bhandari, Kumar & Singh, 2012; Esch et al., 2014; Aleksandrowicz et al., 2014; Gandhi et al., 2015; Nagendra et al., 2015) as well as in those studies successfully combining NDVI with Tasselled Cap (e.g., Chen & Rao, 2008).

Introducing too many variables into a model does not necessarily lead to achieving better results (Lu & Weng, 2007), which underlines the importance of selecting the most appropriate variables for change detection. In this case, the best accuracy was achieved by using directly bands of Landsat image instead of calculated models due to almost all variables (outside the spatial variables) were based on similar spectral bands.

The study results could have been, theoretically, influenced by a number of uncertainties that we, however, strived to eliminate, e.g., through the pre-processing of the satellite images (atmospheric correction, registration of images and its visual verification). No object is shifted by more than 1∕2 a pixel between two frames (Dai, 1998). The selection of the Landsat 8 OLI pairs was predominantly limited by the launch of the satellite mission (2013) and by the cloud cover. Still, a suitable pair of pictures in a suitable phenological phase according to the recommendations (Coppin et al., 2004; Hájková et al., 2012; Esch et al., 2014; Tarantino et al., 2016) was found. The selection of the suitable acquisition period depends on the geographical conditions (especially longitude, latitude or altitude) of the observed area. From this point of view, the presented methods and results are relevant for similar environmental conditions in central Europe. Another uncertainty is a possible error in the LPIS reference database as the land use data is entered directly by the farmers themselves. Also, the information in the LPIS differs slightly from the date of acquisition of the satellite imagery, as it refers to the end of the particular year. No better reference database covering the entire territory of the Czech Republic on such a detailed scale is available however. Moreover, using such a high number of individual plots combined with suitable statistical methods ensured that even if the information was inaccurate by a small fraction, it should not have any significant impact on the results of our study. The accuracy of the resulting change maps could have been affected by selection of the change detection method also. An object-based classification was used in the multi-temporal change detection as it is, according to literature, a more suitable approach for high resolution data, when the pixels are significantly smaller than the object. In this case, grouping pixels into segments is needed (Blaschke, 2010). The ratio of change to no-change units in our study is approximately 1:50 and, therefore, the stratified random sampling design with a proportion of 1:4 (change vs. no-change) for the training and validation data was used.

LULC change detection most commonly employs Post-Classification Comparison (PCC) (Otukei & Blaschke, 2010), it is, therefore, rather a classification than a pure change detection task. For many applications, it is important to describe the trajectory of the change. On the other hand, the knowledge about the occurrence of (no-)change (so-called pre-classification, or bi-temporal change detection) (Coppin et al., 2004) is sufficient for many other tasks. If this is the case, the choice of suitable variables is the key to acquiring quality results, and this is where the contribution of our study can be deemed significant. The methods used here can be applied to CDs of other LULC categories as well. It is a well-known fact that finding suitable variables streamlines analyses, while at the same time improves the results (Lu, Li & Moran, 2014).

Our results indicate that we are nearing a maximum accuracy of the grassland to cropland change detection achievable from a pair of high resolution multi-spectral images. Possible improvements could be brought about by implementing new data into the models. Examples of such supplementary data could include a time series of high resolution images, e.g., Landsat or Sentinel-2 (Esch et al., 2014), very high resolution data (Tarantino et al., 2016), data with a different resolution (Lu, Batistella & Moran, 2008; Turker & Ozdarici, 2011), data captured by other RS methods (Smith & Buckley, 2011), for example radar (Sentinel-1) and thermal data (Landsat 8 TIRS) or the incorporation of an existing GIS database (Hussain et al., 2013). Hussain et al. (2013) and Lu et al. (2003) both state that hybrid methods of change detection combining multiple approaches can increase the accuracy of change detection.

The variables selected in this study can be used with sufficient precision as a source of data for updating existing LULC databases or as a tool for setting agricultural subsidy policies and their implementation. As the reference dataset used in the presented study was quite large, it is relatively safe to assume the applicability of using the results for other studies addressing this change detection problem in the whole of Central Europe. The results are relevant for areas with similar geographical conditions, especially regarding the latitude. However, the selected statistical methods and classification algorithms should be robust due to the fact that the used images (full scene of Landsat 8) covered a large area with topographical variable conditions (lowlands, highlands, mountains).

Conclusions

This study provides an analysis of the utilisation of selected remote sensing variables (vegetation indices, textures, Principal Component Analysis, and Tasselled Cap analysis) for grassland to cropland change detection based on a pair of Landsat 8 OLI images and the Land Parcel Identification System (LPIS) vector database. The results confirm the principal hypotheses that (1) there are suitable variables usable for grassland to cropland change detection; (2) increasing the number of variables used in a model leads to increased accuracy of the change detection, but to achieve the highest accuracy, it is necessary to use original Landsat 8 bands; (3) spectral variables play a more important role than textural variables in the change detection; (4) the appropriate time of the acquisition satellite images is important for grassland to cropland change detection. In view of the accuracy of the created change maps, which was verified using the reference database, we consider a model utilising three variables (namely NDVI, Wetness and Fifth components) the most suitable. Incorporation of additional variables into the model does not significantly improve the accuracy of the change map. By analogy, the methods used in this study can be applied for the CD of other LULC categories than solely those based on grassland to cropland change. The models prepared in this way can serve as data sources for updating the current LULC databases or as a tool for creating agricultural subsidy policies. As the selection of variables was based on a large dataset of reference data on grassland to cropland change detection, the applicability for other studies can be safely assumed. Our conclusions are valid for analyses on a regional scale in Central Europe using high resolution data. To further improve the grassland to cropland change detection using RS, research combining our variables with spatial data acquired using other RS techniques is needed.

Supplemental Information

Supplemental Information 1 Correlation matrices of 59 input variables

Click here for additional data file.

Supplemental Information 2 Summary of calculated models for the grassland to cropland change detection based on different sets of variables

The value of AIC specifies the information potential of the model. Models verified in the study by object-based classification are highlighted in grey.

Click here for additional data file.

Supplemental Information 3 Detailed confusion matrices based on SVM classifier and validation data for models based on one, three, five, seven and fourteen variables and Landsat bands

PA, Producer’s accuracy; UA, User’s accuracy; O, Omission and C, Commission. Change, No change and Sum values represent number of objects. PA, UA, O and C are in percent (%).

Click here for additional data file.

In advance, we acknowledge the anonymous referees for their constructive comments. Thanks to our colleagues from the Department of Applied Geoinformatics and Spatial Planning at the Czech University of Life Sciences Prague (CULS) for their helpful advice.

Additional Information and Declarations

Competing Interests

Author Contributions

Data Availability

The authors declare there are no competing interests.

Tomáš Klouček conceived and designed the experiments, performed the experiments, analyzed the data, contributed reagents/materials/analysis tools, prepared figures and/or tables, authored or reviewed drafts of the paper, approved the final draft.

David Moravec performed the experiments, analyzed the data, contributed reagents/materials/analysis tools, prepared figures and/or tables, authored or reviewed drafts of the paper, approved the final draft.

Jan Komárek analyzed the data, contributed reagents/materials/analysis tools, prepared figures and/or tables, authored or reviewed drafts of the paper, approved the final draft.

Ondřej Lagner analyzed the data, contributed reagents/materials/analysis tools.

Přemysl Štych conceived and designed the experiments, performed the experiments, authored or reviewed drafts of the paper, approved the final draft.

The following information was supplied regarding data availability:

Klouček, Tomáš (2018): Landsat 8. figshare. Dataset. https://doi.org/10.6084/m9.figshare.6322820.v1

Klouček, Tomáš (2018): LPIS_database.zip. figshare. Dataset. https://doi.org/10.6084/m9.figshare.5923567.v1

Klouček, Tomáš (2018): Change maps. figshare. Dataset. https://doi.org/10.6084/m9.figshare.6327035.v1.

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
