# Peer review of "Selecting appropriate variables for detecting grassland to cropland changes using high resolution satellite data"

_PeerJ, doi:10.7717/peerj.5487_

## Round 0.1 · original submission · Major Revisions

The three referees suggest that the submission may be publishable, but only after some major revisions have been made to your manuscript. Therefore, I invite you to respond to their comments and revise your manuscript.

Reviewer 1 ·

Basic reporting

This paper aimed at Grassland-Cropland change detection, it seems the major objective of the paper is to select the optimal feature for the change detection, but the description of the feature importance estimation method is not clear. In addition, the change-detection accuracy is low in my point of view as the UA of change is lower than 10%.The authors should show their novelty and applicability clearly.
Although I am not a native English, I found the language still need some improvement, such as line 240 and 255.

Experimental design

When training the classifier (SVM), only 200 change and 200 no change samples were used, does these training samples enough for training the classifier?

Validity of the findings

As the UA of grass-crop change is low, and the change was significantly overestimated, I donot think the result support the findings well.

Additional comments

1. Line 21, what does the sentence mean? It is not clear
2. Line 65~68, what is the meaning of ‘false one’, I think this sentence should be improved.
3. Line 107~110, I think it is the objective of this study. But it should be like “to find the optimal variable for grass-cropland detection”.
4. Line 174, can you show the correlation matrix?
5. Line 173~179, how do you prove the 41 features are redundant, it is not clear reported.
6. Line 181, can you introduce GLM in detail ?
7. Line 193, you used SVM as classifier, but you should introduce the classifier more clearly. In addition, you should introduce how did you tune the parameter for the classifiers.
8. Why no-change were underestimated, and grass-crop change are signiicantly over estimated?
9. Line 245~246, what’s the sentence mean?
10. It seems the UA of ‘change land’ is too low.
11. Figure 5, you just showed four result maps, you should show the 2013 and 2016 image here.

Reviewer 2 ·

Basic reporting

no comment

Experimental design

The study tested many spectral and spatial variables for change detection. The overall quality of the manuscript is good. There is still room for improvement on how to deal with unbalanced dataset. It is important to use unbiased estimators for accuracy assessment and calculate standard deviation and 95% confidence interval. Blue lines and confidence intervals in Figure 4 do not make sense. Instead, the confidence interval should look like a boxplot with the unbiased estimation in the middle of it, for each selection of variable number. Please re-do accuracy assessment and provide at least one confusion matrix in the response for checking.

Please present at least one resulted map for the whole study area so people can compare it with the reference to see if the distribution of changed areas is similar. This is important for a very unbalanced classification problem.

For the simplest model with only three variables, it should be very helpful to make a 3D scatter plot to demonstrate the separability between the change and non-change classes.

Line 91: texture metrics should be spatial variables.

Validity of the findings

no comment

·

Basic reporting

Title: I suggest to remove the first part of the title (before the colon) which is redundant.

There is some confusion in terminology such as Land Use VS Land Cover (e.g., line 42, grassland and cropland are land cover types) indicating that the authors may not have a complete mastery of the concepts they use. Another example is the reference to Blaschke, 2010 to support the claim that ‘a per-pixel classification … is … a more suitable approach for the high resolution data than the object classification’ (line 288) while you can read in the paper of Blaschke (actually in favor of object-based approach): ‘high resolution: pixels are significantly smaller than object, regionalisation of pixels into groups of pixels and finally objects is needed.’ (see legend Fig. 1).

Figure 2:
Define ‘second moment’. NDVI and second-moment plots are unreadable because of poor contrast and the overlapping polygons. I suggest adding a plot with LPIS polygons at the left side of the figure and removing overlapping polygons to improve readability. You may also want to add a color legend on NDVI and second-moment plots.

English:
I am not a native speaker. Therefore, I understand that is not necessarily easy to write in perfect English. However, I think that the English language of this paper can be largely improved to ensure a smooth reading of the text. Furthermore, in several places in the text, the quality of the language prevents the reader from correctly understanding what the authors mean. I suggest that you request a native English speaking colleagues to review the manuscript.
Examples:
Line 48: ‘helps with the accumulation of greenhouse gases’
Line 75: ‘many papers reviewing’ -> ‘many papers have reviewed’ (try to avoid putting the verb at the end of the sentence).
Line 94: ‘variables of’, end of the sentence is missing.

Structure:
Some tables with results are presented in ‘Materials and Methods’ section in place of ‘Results’ section. Examples: Table A, Table 3.
Line 296: I suggest moving this paragraph to the introduction.

Experimental design

Major issues:
Line 159: This section is essential for the understanding of the paper but poorly described. Please detail the variables used, Table 1 is not sufficient. As an example, what does ‘simple ratio’ means? It could be the ratio of any band… Furthermore, the authors should explain the rationale behind the choice of these variables/indices. Some of them are not obvious regarding cropland and grassland discrimination (e.g. Normalized Difference Built Up Index) or Landsat images (e.g. WorldView Improved Vegetative Index). As it is, it looks like the authors have tested a lot of random indices without prior consideration on the relevance for their objective. This section has to be seriously improved before publication.

Minor issues:
Line 107: Please explain the rationale behind the assumptions described.
Line 124: According to USDA (https://ipad.fas.usda.gov/rssiws/al/crop_calendar/europe.aspx), in August, half of the main crops in the Czeh Republic are still not harvested. Please discuss.
Line 144: FYI, surface reflectance products exist for Landsat 8 and are available at https://lta.cr.usgs.gov/L8Level2SR
Line 155: Please discuss this limitation.
Line 166: Not clear. I guess you mean ‘for each field’.
Line 183: Not clear how you have selected the number of variables.
Line 193: Please provide the parameters of the SVM (kernel…).
Line 195: Why did you use the central pixel of the LPIS unit? I do not understand. Here, you have two options either working by object (pre- or post-classification) or by pixel and then compare all classified pixels with the reference (similarly, the discussion line 292 should be revised).
Line 198: Figure 3 should appear earlier (at the beginning of the Method section).

Discussion: discussion about the added value of temporal series and higher resolution such as Sentinel 2 data (and Radar data) would be interesting.

Validity of the findings

Major issue:
Results and Discussion about the confusion matrix miss the most important point. Change user’s accuracy is, for the best model, around 7% which is not mentioned/discussed in the text. Looking at the confusion matrix of the ‘best model’, 4735 units are classified as change while only 458 are real change + out of these 458 only 350 are actually correctly classified as change and included in the 4735 units. For change detection (the purpose of the study as stated in the title), the proposed method performs poorly (which is not surprising using only one date of data for each year). And you can see it clearly in Figure 4 by comparing the ‘best model’ with the validation data. This point is very important and completely misunderstood in the discussion line 259.

Additional comments

To assess the validity of the findings the method should be more explicit and better described. It seems that the authors have used some kind of feature selection based on logit regression (using the 658 change + 33191 no-change units ? Object-based (zonal stats)?) and then a SVM classification (using 200 change + 200 no-change units, pixel-based) but it is not easy to understand.
Second, the authors have to put their results into the perspective of change detection as explained above. The author may refer to the work of Olofsson et al. (2014) for the best practices for assessing the accuracy of land change.

Reference:
Olofsson, Pontus, et al. "Good practices for estimating area and assessing accuracy of land change." Remote Sensing of Environment 148 (2014): 42-57.

---

## Round 0.2 · Minor Revisions

Reviewers 1 and 3 provided additional comments, please clarify. In particular, the methodology part.

Reviewer 1 ·

Basic reporting

It seems that the paper has improved significantly.

Experimental design

The experimental design of the revised version is fine.

Validity of the findings

The results support the findings of the paper.

Additional comments

The authors used images in August in 2013 and 2016 for change detection and also claimed that "appropriate time of the acquisition of satellite images is important for grassland to cropland change detection". But, in other study regions, the growing season of grass and crops may vary, the selected time phases for grassland-to-cropland change detection will be different from this paper. I suggest the authors address this issue.

Reviewer 2 ·

Basic reporting

no comment

Experimental design

no comment

Validity of the findings

no comment

Additional comments

Issues have been addressed and I have no further comment.

·

Basic reporting

While efforts have been made to improve the paper compared to the first version, it is still poorly written and sometimes hard to understand. Several statements are made without being supported by any scientific evidence or reference.

In particular, the authors failed to address one of the major issue raised in my original review regarding the section "2.4. Selection and calculation of the variables" (line 179). The only explanation found in the text to support the choice of the considered variables is that "these [...] variables represent, in our opinion, all potentially used spectral and spatial indicators for change detection in the ENVI software by a common user" (line 182). This a poor justification to support what is supposed to be the main focus of the paper (see the title: "Selecting appropriate variables..."). Furthermore, the authors overstate several times what has been done by claiming to "comprehensively address the effect of spectral and textural variable" (line 114) and to "provide a complex analysis of the utilization of remote sensing variables" (line 353). Both statements are false. The theoretical background that led to the choice of the feature selection method (AIC comparison of logistic regression) is also omitted.

Regarding the results, if I understood well (not sufficiently clear in the paper), it seems that using the Landsat bands alone give the best results for the SVM classification ("to achieve the highest accuracy, it is necessary to use original Landsat 8 bands" (line 358 + Table 4, Figure 5)). If correct, this has major implications for this study (so far as to question its purpose) which are completely omitted by the authors.

In the conclusions, the authors claim that "the results confirm the principal hypotheses that [...] the appropriate time of the acquisition satellite images is important for grassland to cropland change detection". While it is, indeed, a significant limitation of this work, I was not able to found where it was discussed in the paper.


Minor issue:
line 212: What are the implications of using > 50% of plots w/ change and ~3% of plots w/o change for training data?
line 269: What are “green” crops?
Figure 3: convolution matrix should be confusion matrix

Experimental design

-

Validity of the findings

-

---

## Round 0.3 · accepted · Accept

I am pleased to inform you that your paper has been accepted for publication.

#